# Strategies for Improved Wettability of Polyetheretherketone (PEEK) Polymers by Non-Equilibrium Plasma Treatment

**DOI:** 10.3390/polym14235319

**Published:** 2022-12-05

**Authors:** Gregor Primc

**Affiliations:** Department of Surface Engineering, Jozef Stefan Institute, Jamova cesta 39, 1000 Ljubljana, Slovenia; gregor.primc@ijs.si

**Keywords:** non-equilibrium gaseous plasma, PEEK, functionalization, wettability

## Abstract

Polyetheretherketone (PEEK) is the material of choice in several applications ranging from the automotive industry to medicine, but the surface properties are usually not adequate. A standard method for tailoring surface properties is the application of gaseous plasma. The surface finish depends enormously on the processing parameters. This article presents a review of strategies adapted for improved wettability and adhesion of PEEK. The kinetics of positively charged ions, neutral reactive plasma species, and vacuum ultraviolet radiation on the surface finish are analyzed, and synergies are stressed where appropriate. The reviewed articles are critically assessed regarding the plasma and surface kinetics, and the surface mechanisms are illustrated. The directions for obtaining optimal surface finish are provided together with the scientific explanation of the limitations of various approaches. Super-hydrophilic surface finish is achievable by treatment with a large dose of vacuum ultraviolet radiation in the presence of oxidizing gas. Bombardment with positively charged ions of kinetic energy between about 100 and 1000 eV also enable high wettability, but one should be aware of excessive heating when using the ions.

## 1. Introduction

Polyetheretherketone (PEEK) is a thermoplastic polymer with excellent mechanical and electrical properties and chemical stability even at temperatures up to 250 °C and above. These properties make PEEK an attractive material for application in extreme conditions and thus suitable as an insulation material in automotive and aerospace industries [1]. Furthermore, the properties also make PEEK promising for use in medicine, for example, as a material for dental implants [2,3]. As known for most polymers, the mechanical properties are further improved by enforcement with different fillers, often in the form of fibers. The excellent chemical stability, however, may be a drawback in numerous applications where PEEK should be applied as a coating or should be coated with another material.

The poor adhesion properties are due to the PEEK’s chemical structure, which prevents interaction with most other materials. It is a moderately hydrophobic polymer with a static water contact angle (WCA) of about 90°. The interaction of all polymers with foreign material is improved by grafting various functional groups on its surface. The grafting may result in improved wettability, which is usually desired before coating with various materials, such as metallization, gluing, printing, or decorating with nanomaterials. Grafting with functional groups may not be suitable when PEEK in a liquid state is to adhere in the form of a thin film to another material. In such cases, the material coated with a thin PEEK film should be pre-treated to ensure improved adhesion, and thus, adequate mechanical properties of the product.

A review of the strategies for improving the adhesive properties of PEEK by the treatment with non-equilibrium gaseous plasma is the topic of this article. First, the literature on surface modification of solid PEEK by gaseous plasmas is reviewed and discussed, and second, the surface modifications using various approaches are illustrated. Lastly, the missing results are recognized, and future research is recommended.

## 2. Grafting of Functional Groups on the PEEK Surface and Morphological Changes

As early as 1998, Inagaki et al. [4] reported the surface functionalization of PEEK using oxygen plasma. Oxygen plasma is a natural choice for the surface functionalization of fluorine-free polymers [5]. The reactive species formed in oxygen upon plasma conditions interact chemically with the polymer surface, causing either a direct substitution of H atoms bonded to carbon with oxygen or bond breakage. The dangling bonds are occupied with oxygen atoms, thus forming a variety of oxygen-containing functional groups such as hydroxyl, epoxy, carbonyl, carboxyl, etc. [6]. Theoretically, these surface functional groups should ensure an increase in the polar component of the surface energy and, hence, an increase in the wettability. The increased wettability should ensure good coating adhesion on the PEEK’s surface or adhesion between two PEEK samples. The exact mechanisms of interaction between oxygen plasma and polymer surface on the atomic scale are yet to be elaborated. Among recent spotless theories, Ventzek’s group elaborated on the kinetics of surface functionalization of polystyrene [7], and the results were confirmed by a carefully designed experiment [8]. No suitable theory has been reported for PEEK, though. On the contrary, Inagaki et al. [4] reported a rather poor functionalization but significant modifications of the PEEK surface film. Namely, the oxygen plasma treatment caused the formation of low molecular weight fragments on the PEEK’s surface. The fragments were easily washed away, and the resulting WCA observed after washing was not improved dramatically. Inagaki et al. used two configurations for plasma treatment, i.e., the glowing plasma, sustained by an electrodeless radiofrequency (RF) discharge in the capacitive mode, and the flowing afterglow. The difference between these two configurations is in the reactive species available for surface modification of a polymer. Afterglow consists of neutral oxygen atoms in the ground state and some metastable neutral molecules, whereas the glowing plasma also consists of charged particles and is a significant source of radiation [9]. The most important for polymer surface modification is radiation in the vacuum ultraviolet (VUV) part of the spectrum [10]. VUV photons have energies well above the binding energy of atoms in polymers and thus create dangling bonds in the surface film.

Inagaki et al. [4] presented plots of the WCA versus the treatment time between 5 and 120 s and discharge powers from 20 and 60 W. In all cases, the WCA dropped significantly after the shortest treatment time and remained unchanged thereafter. When treated in the glowing plasma, PEEK exhibited a WCA of about 17° at the lowest discharge power of 20 W and 9° at the largest power of 60 W. After washing, oxygen-plasma-treated PEEK exhibited a WCA of 62° and 70° for samples treated at the powers of 20 and 60 W, respectively. Therefore, the larger discharge power caused poorer wettability after washing away the molecular fragments. Even more contradicting results were obtained in the afterglow, where the WCA measured on washed samples was 44° and 77° for samples treated at the powers of 10 and 60 W, respectively. Some samples were mounted in a box with an MgF_2_ window to test the influence of VUV radiation on the surface wettability. Even a minute of treatment with VUV radiation arising from oxygen plasma did not cause any detectable modification of PEEK’s wettability.

The surface structure was determined from high-resolution X-ray photoelectron spectra (XPS) by Inagaki et al. [4]. The deconvolution of the C1s and O1s peaks led to the conclusion that the treatment in the glowing plasma did not change the surface structure much, but the afterglow treatment caused an increased concentration of both C−O and C=O groups, wherein the increase was twofold for the C−O group and less for the C=O group. The satellite peak characteristic for aromatic polymers vanished after treating PEEK both in the glowing plasma and afterglow, indicating the destruction of the aromatic ring in the surface film probed by XPS. The [O]/[C] concentration after treatments with glowing plasma or afterglow increased from 0.13 to about 0.20. The concentration of highly polar groups, such as O−C=O, was below the detection limit of XPS. Based on the results reported by Inagaki et al. [4], one can conclude that the weakly ionized, highly dissociated oxygen plasma causes preferential degradation and etching of the PEEK surface film rather than functionalization with highly polar functional groups. The effect is illustrated in Figure 1.

Results similar to those of Inagaki et al. [4] were reported by Narushima and Ikeji [11], except that they used a temporal afterglow instead of the flowing afterglow. The temporal afterglow was obtained by pulsing the RF generator operating at the frequency of 13.56 MHz and the power of 50 W. The oxygen pressure was set to 13 Pa. The plasma pulses lasted 10 µs and were repeated every millisecond. The authors stated that the electrons began to decrease at about 100 µs whereas the oxygen radicals at about 100 ms. Taking into account these values, the density of neutral radicals was almost constant over the treatment time of 1 min, but the charged particles and VUV radiation were applied in pulses. The surface finish, as reported by Narushima and Ikeji [11], was independent of the treatment mode (continuous or pulsed). In both cases, the concentration of C−O functional groups as deduced from XPS spectra increased from 20 to 25%, and C=O from 5 to 6%. No other functional groups were found on the PEEK’s surface after either of the plasma treatments. The [O]/[C] ratio, as deduced from XPS survey spectra, increased from 0.16 for untreated samples to 0.23 for plasma-treated samples. The difference between the continuous and pulsed modes was in the etching rate, which was found to be 5 and 1 µg/cm^−2^ min^−1^ for continuous and pulsed treatments, respectively. The WCA observed after the treatments were 66° in the case of the continuous mode and 62° for pulsed mode. The pulsed mode, therefore, somehow enabled better wettability than the continuous mode, but neither were sufficient to achieve a very low water contact angle. Interestingly enough, the adhesion force between deposited copper and the plasma-treated PEEK samples was better in the continuous mode, at 800 mN cm^−2^, whereas in the pulsed mode, it was 360 mN cm^−2^. The adhesion force was much lower, 40 mN cm^−2^ for the untreated PEEK samples. The oxygen plasma treatment, therefore, assured an order of magnitude larger adhesion force.

More recently, Han et al. [12] also probed oxygen plasma for the surface activation of PEEK samples for application in dentistry. The samples were prepared by 3D printing and exposed to oxygen plasma. Plasma was sustained by an RF discharge at 40 kHz, and the pressure and power were 100 Pa and 100 W, respectively. The treatment time was 15 min and the WCA 25 and 38° for the as-printed and polished samples, respectively. The authors did not mention any washing, but the hydrophobic recovery caused an increased WCA to about 60° after 3 weeks of aging. The surface finish was, therefore, similar to that shown in Figure 1.

The observation of Inagaki et al. [4] of PEEK’s surface wettability is not in agreement with the recent paper by Arikan et al. [13]. Arikan and co-workers performed a systematic study of the surface kinetics versus the fluence of VUV radiation and found rich surface chemistry. The [O]/[C] concentration after treatment with VUV radiation increased to over 0.3 even after receiving a moderate fluence of VUV radiation of about 1 J/cm^2^. Furthermore, Arikan et al. [13] also observed both COOH and COOR groups on the surface of VUV-treated PEEK. The optimal adhesive strength on VUV-treated PEEK was obtained already after 5 s irradiation with VUV radiation.

Arikan et al. [13] used an Xe excimer lamp that emits radiation in a broad wavelength range from about 150 to 190 nm, peaking at the wavelength of 172 nm (corresponding photon energy 7.2 eV). The flux of VUV radiation was adjusted by changing the treatment time from 3 to 720 s, corresponding to the photon doses between 5 × 10^16^ and 13 × 10^18^ cm^−2^. The corresponding energy absorbed by the PEEK samples due to VUV photon absorption was between 57 and 14,500 mJ cm^−2^. The samples were placed in the ambient air next to the VUV lamp. Upon these conditions, excellent hydrophilization was achieved. The authors did not report WCAs, but mentioned that the surface energy of the PEEK samples treated even at moderate VUV fluences was as high as 70 mJ m^−2^, which was close to the detection limit using water droplets. The polar component was about 25 mJ m^−2^ for samples treated by VUV radiation, while the dispersive component remained practically unchanged at about 45 mJ m^−2^.

The authors also estimated the etching rate due to VUV irradiation and found rather linear behavior at the rate of about 7 nm J^−1^ cm^2^ [13]. The etching is explained by the formation of dangling bonds in the PEEK surface film. The dangling bonds interact with molecular oxygen (since the samples were kept in the air in close proximity to the VUV lamp) during the treatment. The etching was accompanied by nanostructuring of the surface. Atomic force microscopy (AFM) revealed the appearance of well-defined periodical structures of circular shape and a typical lateral dimension below 100 nm. The nanostructured surface already appeared at moderate VUV fluences (190 mJ cm^−2^) and did not change much with prolonged treatment since the AFM image at 190 mJ cm^−2^ was practically identical.

XPS characterization revealed a gradual increase in the [O]/[C] ratio, which peaked at the VUV fluence of about 1000 mJ cm^−2^. It slowly decreased with prolonged treatment time, but remained over 0.3 even at the largest fluence of 14,000 mJ cm^−2^. The maximal [O]/[C] ratio was about 0.37 for amorphous and 0.33 for semi-crystalline PEEK. Unlike Inagaki et al. [4], Arikan et al. [13] also observed highly polar functional groups on the PEEK surface after VUV treatments. The concentration of various polar groups increased between 190 and 6840 mJ cm^−2^, with the concentration of O−C=O groups deduced from the high-resolution XPS C1s peak as large as 10% at the largest VUV fluence.

The almost optimal surface finish of VUV-treated PEEK samples resulted in an excellent adhesive bond strength. While the strength of untreated samples was only 3 and 5 MPa for semi-crystalline and amorphous PEEK, respectively, it rose to about 20 MPa for both samples after being exposed to the VUV fluence of about 100 mJ cm^−2^, and reached about 25 MPa at the fluence of about 1000 mJ cm^−2^. The VUV irradiation of PEEK samples thus not only causes excellent wettability but adhesive bond strength as well. The surface effects upon treatment of PEEK by 172 nm VUV radiation is illustrated in Figure 2. The pristine samples are exposed to a rather intensive VUV radiation and simultaneously to air. Air at ambient conditions consists of stable molecules capable of chemical interaction with dangling bonds formed on the PEEK’s surface upon VUV irradiation. No nitrogen was observed on the PEEK’s surface, so the dangling bonds indeed interacted only with oxygen-containing molecules. The air moisture was not reported by Arikan et al. [13], so it would not be justified to exclude interaction with water vapor. Still, the oxygen content in the air, even at large humidity and room temperature, is much larger than the concentration of water molecules, so it is reasonable to assume the interaction of O_2_ with dangling bonds. The interaction causes the formation of various oxygen-containing functional groups, while prolonged irradiation causes extensive etching and desorption of interaction products such as carbon oxides and water vapor. The net effect after the prolonged treatment is thinning the PEEK samples without a significant effect on surface morphology or chemical structure. The combination of highly polar surface functional groups and nanostructured polymer morphology always results in a super-hydrophilic surface finish [6]. The irradiation with VUV photons from the Xe excimer lamp on the PEEK surface finish was also elaborated in [14].

Modification of the PEEK surface properties by VUV radiation was also reported in a recent paper by Yoshida et al. [15]. They used a lamp emitting light at 185 or 254 nm wavelengths (probably a mercury lamp) operating at the power of 200 W. The PEEK samples were placed in an oxygen atmosphere at a pressure of 20 mbar and about 140 mm away from the lamp. The irradiation times were 1, 5, and 10 min, but the data on the photon fluxes are not available. At such conditions, Yoshida et al. [15] reported a roughly exponential decrease in the WCA with the treatment time, but the WCA after 5 min irradiation was still about 30°. The discrepancy with results reported by Arikan et al. [13] could be explained by the different wavelengths used by the authors. Another explanation could be different doses of photons, but the dose was only reported by Arikan et al. [13].

Extreme UV radiation was used for the surface modification of PEEK samples by Czwartos [16]. The EUV was supplied in pulses and the intensity was large enough to cause the formation of plasma above the polymer surface. The initial WCA was 74° and it dropped to 64° after 200 pulses of EUV irradiation. The XPS characterization revealed significant modifications of the surface composition and formation of various functional groups. The discrepancy between rich functionalization and a rather marginal increase in wettability may be explained by thermal effects.

An alternative approach to bond scission in the surface film of PEEK polymers by absorption of VUV radiation is the application of ion beams. High-energy ions cause radiation damage over a rather thick surface film and also significant heating, so they were found inappropriate by Kim et al. [17]. Instead, Kim et al. used relatively low-energy ion beams (below 1 keV). The penetration depth of such ions in PEEK was estimated to be about 8 nm using the TRIM96 code. The source of Ar^+^ ions was a hollow cathode discharge, and the ion fluence varied between 1.6 × 10^18^ and 1.3 × 10^20^ cm^−2^. The ion fluences used by Kim et al. [17] were therefore similar to the VUV fluence used by Arikan et al. [13]. Oxygen was introduced in the vacuum chamber at a low flow rate of a few sccm to ensure vacuum conditions upon treatment of PEEK samples. Such a low pressure suppressed the loss of Ar^+^ ions’ kinetic energy at elastic collisions with neutral gas in the processing chamber.

Kim et al. [17] monitored the surface chemistry by XPS. The irradiation with Ar^+^ ions in the absence of oxygen enabled a gradual increase in the [O]/[C] ratio, from an initial 0.13 to 0.16 after the irradiation time of 30 s, and 0.22 after the irradiation time of 120 s. The surface functionalization in the irradiation chamber without introducing oxygen may be explained by the residual atmosphere, which usually consists of water vapor in hermetically tight vacuum systems. The ion irradiation causes bond scission in the surface film, and the dangling bonds interact with H_2_O to form oxygen functional groups. The base pressure in the reaction chamber during treatment with Ar^+^ ions was about 0.01 Pa, so the flux of molecules in the residual atmosphere was still as large as about 3 × 10^20^ m^−2^s^−1^. The deconvolution of the high-resolution C1s peak revealed the appearance of the O−C=O groups on the PEEK’s surface. The resulting WCA after irradiation with Ar^+^ ions was about 70°, so the hydrophilicity was slightly improved.

Kim et al. [17] reported even better hydrophilization upon treatment with ions and the presence of oxygen in the reaction chamber. In this case, the WCA of about 50° was obtained already at the treatment time of about a minute. Prolonged treatment did not cause any further decrease in the WCA. The [O]/[C] ratio of 0.28 was obtained at irradiation in the oxygen atmosphere at low pressure, and the concentration of the O−C=O groups on the PEEK’s surface was about 4%. Copper was deposited on treated samples, and the shear strength increased from 10 to 60 mN cm^−2^ at the treatment time of about 2 min. Prolonged treatment resulted in somewhat lower shear strength, which may be explained by heating of the PEEK’s surface due to Ar^+^ bombardment. Etching was not mentioned by Kim et al., but it is expected that the treatment causes the removal of the oxidized surface layer due to the combined effects of surface oxidation and ion bombardment.

The method introduced by Kim et al. [17], therefore, enabled moderate hydrophilicity of PEEK surfaces. It is better than treatment with oxygen plasma, as reported by Inagaki et al. [4] and Narushima and Ikeji [11], but not as effective as irradiation with VUV radiation, as reported by Arikan et al. [13]. Both energetic Ar^+^ ions and VUV radiation from the Xe excimer lamp are capable of breaking bonds in the surface film, but the energy of ions is much larger than the photon energy, so the thermal effects cannot be excluded when using Kim’s method, which is illustrated in Figure 3.

Energetic ion beams were also used by Awaja et al. [18] to improve the adhesion of biological cells on the PEEK surface. They used ions created in plasma sustained in O_2_ and CF_4_ to bombard the polymer surface with ions having kinetic energy between 2 and 20 keV. Such energetic ions caused etching and functionalization, but the minimum WCA was about 40°. The cell adhesion increased with decreasing WCA. The rather inadequate wettability could be explained by thermal effects because other authors reported a super-hydrophilic surface finish upon irradiation with energetic ions [19]. Actually, cell adhesion was found to be optimal in the limited range of ion energies and treatment time. The best results were observed at the bias voltage of 6 kV and the treatment time of about 100 s.

More recently, Kruse et al. [20] also reported improved adhesion of biological cells on PEEK substrates using the plasma immersion ion implantation technique. Energetic nitrogen ions bombarded the polymer surface in pulses of duration 45 µs with a repetition frequency of 1500 Hz. The ion kinetic energy was 10 keV. A super-hydrophilic surface finish was reported for the treatment time of 20 min. The observations reported in [20] clearly illustrate the need to prevent overheating of the polymer surface upon ion bombardment. Treatment with energetic nitrogen ions was also reported by Zheng et al. [21], while Zhao et al. [22] used the plasma immersion ion-implantation technique with water vapor and ammonia plasmas. Biocompatible PEEK properties are also improved by the implantation of metallic ions [23].

A variety of gases were used for sustaining plasmas suitable for modification of the PEEK’s surface [2]. Prolonged treatment of PEEK with oxygen plasma was reported by Botel et al. [24]. Plasma was sustained in oxygen or a mixture of oxygen and argon at the pressure of 30 Pa with a capacitively coupled discharge powered by a 100 kHz RF generator at the power of 200 W. The reported temperature of the samples during plasma treatment was 70 °C, and the treatment times were either 3 or 35 min. The authors reported the WCA after the treatment of 0.0° for all samples except those treated for 35 min in Ar/O_2_ plasma, where the WCA was 2.8°. The authors reported a super-hydrophilic surface finish of the PEEK samples. No details about the plasma parameters are disclosed in [24], hence it is difficult to explain the reasons for the super-hydrophilic surface finish. One possible explanation would be bombardment with O_2_^+^ ions if the samples were placed on the powered electrode. Namely, the low-frequency capacitively coupled discharge operates at the voltage of several 100 V. Self-biasing occurs on the surface of samples placed on the powered electrode, so the samples are bombarded with O_2_^+^ (and perhaps also O^+^) ions of several 100 eV kinetic energy. The rather large discharge power—as compared to [4] and [11]—of 200 W may also assure for rather extensive radiation in the VUV range. The synergy between ions and VUV radiation may lead to significant surface structural modifications. Another explanation may be in the appropriate roughness of the samples. Namely, the samples were first polished and then sand-blasted, resulting in a roughness R_a_ of about 0.7 µm. The nanoscale roughness was not reported, but bombardment with energetic oxygen ions should cause a rich morphology on the nanometer scale. It is known that the required conditions for the super-hydrophilic surface finish are both functionalization with highly polar functional groups and rich morphology on the submicrometer scale [6]. A feasible explanation of the surface effects using Botel’s method [24] is illustrated in Figure 4.

Despite the excellent wettability, the shear bond strength between PEEK samples and veneering composites did not change significantly because of plasma treatments. The feasible explanation for this rather unexpected observation is the application of special bonding agents purchased from Visiolink. A possible application of this technique is in dentistry. The same application was addressed by Akay et al. [25], but they used plasma-treated PEEK-PMMA (polymethyl methacrylate) composites.

Dental applications of plasma-treated PEEK were also addressed by Younis et al. [26]. They probed plasmas of various gases, including argon, air, oxygen, and nitrogen. They used a commercial plasma reactor intended for dentistry. Plasma was sustained in a cylindrical tube with an RF generator operating at the frequency of 40 kHz, and the output power was 100 W. No details of the discharge coupling or plasma parameters are reported in [26]. The treatment time was 10 min, and the authors mentioned that the PEEK samples were kept at 20 °C during the plasma treatment. The gas pressure was 30 Pa. The shear bond strength was measured after the plasma treatments. It increased by a factor of two as compared to the untreated PEEK samples and was independent of the type of gas used. The shear strength was almost the same as for glued samples.

Contrary to [26], Fedel et al. [27] also probed argon, oxygen, and nitrogen plasmas, and found nitrogen plasma particularly useful. A capacitively coupled RF discharge was powered with the generator operating at the frequency of 13.56 MHz and the output power of 100 W. The ultimate pressure in the discharge chamber was 1.6 × 10^−2^ Pa, and the working pressure was 0.67 Pa. The treatment time was 2 min. The samples were rather large, with dimensions 10 cm × 10 cm. Nitrogen plasma treatment significantly increased the strength of self-bonding in both crystalline and amorphous samples. The bonding strength was more than doubled when pressing for 4 h and tripled after pressing for 7.5 h at 200 °C. As expected, the plasma treatment did not have a measurable influence on the crystallinity, which was monitored by X-ray diffraction (XRD). The authors explained improved adhesion by surface functionalization and formation of dangling bonds, but did not use a surface-sensitive technique to elaborate the effect.

The surface chemistry upon treatment of PEEK with nitrogen plasma was discussed by Wang et al. [28]. They used a low-temperature plasma reactor powered by an RF generator operating at the frequency of 13.56 MHz and voltage of 500 V. No other details about the discharge system or the nitrogen pressure were disclosed. The treatment times were 15, 25, and 35 min. Scanning electron microscope (SEM) images revealed the evolution of a rich morphology on the submicrometer scale due to the plasma treatment. The XPS survey spectra showed an appearance of nitrogen on PEEK surfaces, but the results were not quantified. The wettability of the as-synthesized PEEK was rather good with the WCA of 73°. Specifically, this value differs from values reported by other authors (just above 90°). The WCA was found to decrease gradually with increasing treatment time: it was 59° after treating for 15 min and 35° after treating for 25 min. The shear bond strength did not follow the evolution of the wettability because it was the highest after 25 min of plasma treatment, when the WCA was 53°. The shear strength was about three times larger than for untreated samples. The authors explained the improved adhesion properties by introducing nitrogen-containing functional groups. They proposed a chemical bond of the surface nitrogen group with the dimethacrylate elements of the Variolink. The proposed interaction between nitrogen plasma and the PEEK surface is illustrated in Figure 5.

More recently, Bres et al. [29] used a powerful plasma torch for surface modification with either air or nitrogen plasma. The samples were placed in the afterglow to prevent melting, since the typical discharge power used for powering the atmospheric plasma torch was 1000 W. The surface tension increased from 53 to about 74 mN/m, and the resultant hydrophilicity depended on the distance between the glowing hot plasma and the sample. Large-impedance atmospheric-pressure discharges may be more suitable for deposition and tailoring surface properties of polymers [30,31], although the treatment uniformity over a large area is still a technological challenge [32].

## 3. Conclusions and Recommendations

This review on the non-equilibrium plasma techniques for surface modification of PEEK polymers explains the observations reported by various authors who probed different plasmas for achieving improved wettability. Plasma provides reactive species and radiation in the UV and VUV range, which are capable of causing both reversible and irreversible modifications. The surface finish depends enormously on the fluxes of reactive species and radiation, and the thermal effects should not be neglected in some cases. The major advance is providing readers with the influence of various species and radiation, which explains large variations in the surface wettability reported by authors who used different plasmas.

The literature survey indicates that the best conditions for improved wettability include treatment with VUV radiation in the presence of oxygen. Such a combination was also suitable for the activation of fluorine-containing polymers such as Teflon [33]. The VUV photons break even the strongest bonds in the polymer surface film [34]. The dangling bonds interact with gaseous radicals and even molecules in the ground electronic state, thus resulting in surface functionalization. The choice of the VUV source is not particularly limited since the radiation at a particular wavelength interacts with the polymer, irrespective of the source. The most common source of VUV radiation is gaseous plasma. Most plasmas irradiate in the VUV range, but the range of wavelengths and intensity depends enormously on the gases used in the plasma sources and the peculiarities of the discharges. Excimer and exciplex lamps operating at atmospheric pressure are sources of extensive VUV radiation. A VUV-transparent window should be placed between the plasma generated by excimer lamps and the PEEK samples.

Alternatively, the polymer samples are placed directly into the gaseous plasma. Most authors used this configuration for treating PEEK samples. The VUV radiation from low-pressure plasma sources useful for surface functionalization of PEEK samples will increase with increasing electron temperature and density, and thus with increasing discharge power (or, rather, power density). The VUV radiation from oxygen plasma arises from the resonant radiative transition of O-atoms [35]. The O-atoms, however, also interact chemically with the PEEK surface, which may lead to excessive chemical etching and the formation of low molecular weight fragments of poor adhesion on the PEEK’s surface. Oxygen plasma is, therefore, useful for the desired surface finish of PEEK, providing the chemical interaction is suppressed by lowering the fluxes of reactive plasma species and favoring the flux of VUV radiation.

Apart from VUV radiation, the bond scission in the surface film of polymers is also achievable by treatment with energetic ions. In this case, one should be aware of surface charging and excessive heating. The surface charging will cause non-uniform treatment, so it is advisable to perform the treatment under the presence of low-energy charged particles, i.e., plasma conditions. The excessive heating can be suppressed by using moderately energetic ions, for example, in the range from 100 to 1000 eV. Such ions are found next to the polymer surface in cases where self-biasing occurs, for example, by placing the polymer samples on the powered electrode of a capacitively coupled RF discharge. Alternatively, it is advisable to apply energetic ions in pulses to prevent excessive heating and, thus, the loss of polar groups due to thermal effects.

Oxygen plasmas with a high density of reactive species (O-atoms in the ground and metastable states and metastable molecules), low VUV radiation, and no ability to accelerate positively charged ions onto the PEEK’s surface are not recommended for functionalization of the PEEK’s surface because of substantial etching. Such plasmas are sustained by electrodeless RF discharges in the continuous mode or in the afterglows (both temporal and flowing). The surface wettability is increased significantly using such plasmas rich in neutral radicals, but low molecular fragments are formed, which may suppress the adhesion properties.

Nitrogen plasma is an attractive alternative to oxygen because it is chemically less reactive than oxygen at a given discharge power. This means that the chemical etching will be lower than in the case of oxygen plasma. Still, chemical etching occurs and causes the formation of hydrogen cyanide, so some precautions are recommended when using plasmas containing nitrogen, including ammonia. 

Finally, it should be stressed that the surface finish depends on the fluxes and fluences of chemically reactive species, VUV radiation, and positively charged ions. While some authors reported the fluxes of VUV radiation and ions, none reported the fluxes of radicals. The authors are therefore encouraged to measure the plasma parameters upon treatment of PEEK samples.

## Figures and Tables

**Figure 1 polymers-14-05319-f001:**
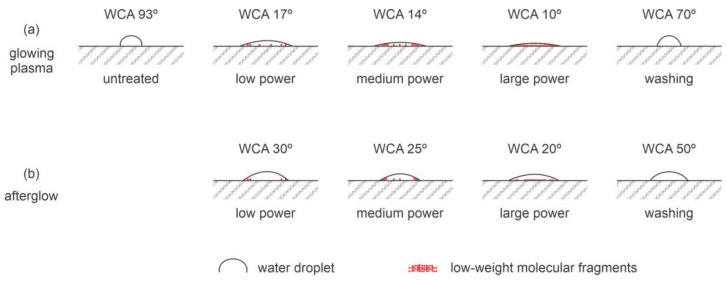
Schematic surface effects during PEEK treatment with weakly ionized oxygen plasma (**a**) and oxygen plasma afterglow (**b**).

**Figure 2 polymers-14-05319-f002:**
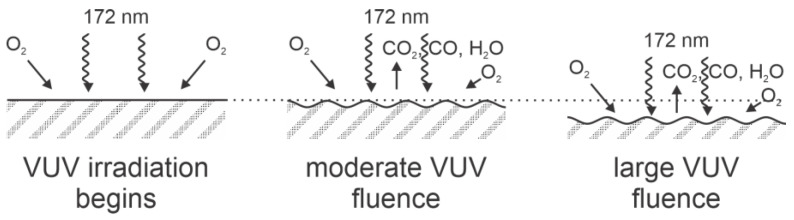
Schematic of the surface effects during PEEK treatment with VUV radiation from an Xe excimer lamp.

**Figure 3 polymers-14-05319-f003:**
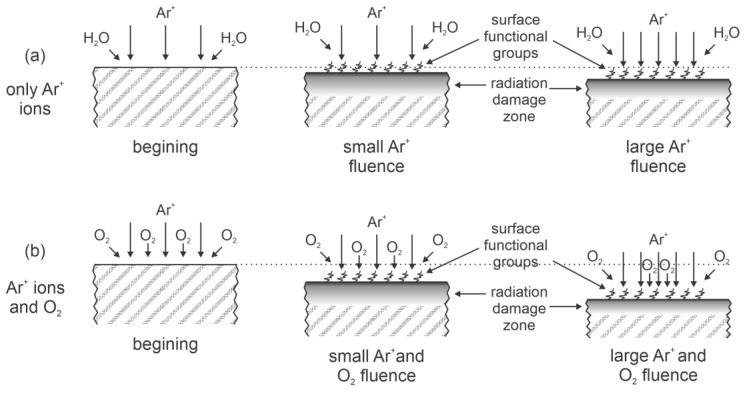
Illustration of the surface effects upon irradiation of PEEK with Ar^+^ ions in the absence of oxygen (**a**) and oxygen presence (**b**).

**Figure 4 polymers-14-05319-f004:**
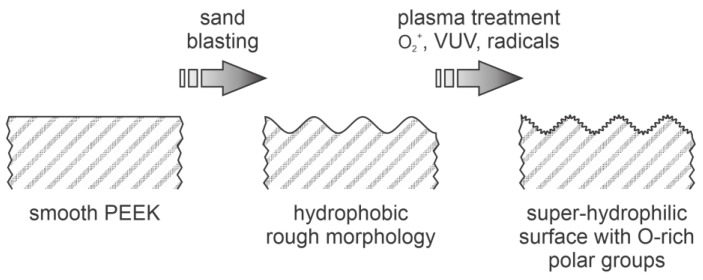
Schematic of the methods for super-hydrophilic surface finish of PEEK samples.

**Figure 5 polymers-14-05319-f005:**
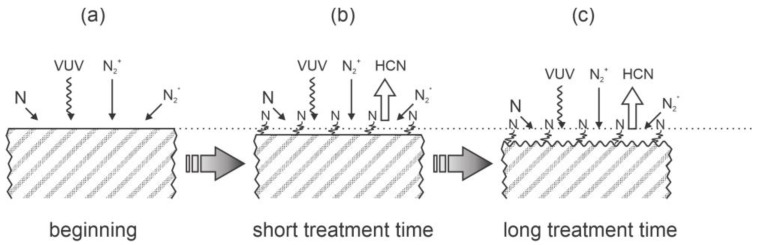
Schematic of the interaction between nitrogen plasma and PEEK surface at the beginning of treatment (**a**), after short (**b**) and long treatment time (**c**).

## Data Availability

Not applicable.

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
