# Peer review of "Strategies for Improved Wettability of Polyetheretherketone (PEEK) Polymers by Non-Equilibrium Plasma Treatment"

_polymers, 2022, doi:10.3390/polym14235319_

Round 1

Reviewer 1 Report

The author focused on a review of the strategies for improving the adhesive properties of PEEK. Although this work shows some interesting or new insights, there are still many problems to be solved. Please see the detailed comments in the below.

(1) In section ‘Abstract’, authors pointed out the wettability which is a little difference compared that to title of this work. Please check it!

(2) In section ‘Introduction’, the content is too short, authors must introduce and explain relevant contents in recent years.

(3) In section ‘Introduction’, Why did the authors only provide non-equilibrium gaseous plasma? How about other ways?

(4) In Conclusion, authors should present a strong case for how this work is a major advance.

Author Response

Dear Reviewer, thank you for your comments and suggestions.

The author focused on a review of the strategies for improving the adhesive properties of PEEK. Although this work shows some interesting or new insights, there are still many problems to be solved. Please see the detailed comments in the below.
(1) In section ‘Abstract’, authors pointed out the wettability which is a little difference compared that to title of this work. Please check it!
Yes, the title might be too general. I changed the title to “Strategies for improved wettability of Polyetheretherketone (PEEK) polymers by non-equilibrium plasma treatment.”
(2) In section ‘Introduction’, the content is too short, authors must introduce and explain relevant contents in recent years.
Thanks for this comment. This is a review paper, so I just presented a brief motivation for why improved wettability is demanded. The relevant publications are analyzed in section 2.
(3) In section ‘Introduction’, Why did the authors only provide non-equilibrium gaseous plasma? How about other ways?
Because I wanted to keep the review as concise as possible. Of course, numerous other methods exist, but the literature is vast, so I focused on non-equilibrium plasma treatment.
(4) In Conclusion, authors should present a strong case for how this work is a major advance.
Thank you for this suggestion. I added the following paragraph at the beginning of section 3 (Conclusions): “This review on the non-equilibrium plasma techniques for surface modification of PEEK polymers explains the observations reported by various authors who probed different plasmas for achieving improved wettability. Plasma provides reactive species and radiation in the UV and VUV, which are capable of causing both reversible and irreversible modifications. The surface finish depends enormously on the fluxes of reactive species and radiation, and the thermal effects should not be neglected in some cases. The major advance is providing readers with the influence of various species and radiation, which explains large variations of the surface wettability reported by authors who used different plasmas.”

Kind regards

Reviewer 2 Report

Purely review work. Nevertheless, correct selection of literature items and proper interpretations of the authors' results. An interesting article from the point of view of the issues discussed in one place. I have comments about the distance between the drawings and the text above the drawings - the distance could be greater. In addition, the article is properly formatted, written in correct language. The notations and units used are also spelled correctly. The drawings are legible and with the correct resolution. Self-citations in an acceptable amount. The work is suitable for publications after minor corrections.

Author Response

Dear Reviewer, thank you for your comments and suggestions.

Purely review work. Nevertheless, correct selection of literature items and proper interpretations of the authors' results. An interesting article from the point of view of the issues discussed in one place. I have comments about the distance between the drawings and the text above the drawings - the distance could be greater. In addition, the article is properly formatted, written in correct language. The notations and units used are also spelled correctly. The drawings are legible and with the correct resolution. Self-citations in an acceptable amount. The work is suitable for publications after minor corrections.

Thanks for spotting the inadequate distance between the text and the drawings.

Kind regards

Reviewer 3 Report

The article presents an overview of modern strategies to improve PEEK wettability and adhesion. Although the review is up-to-date, here are some comments on it:

1) The abstract should be expanded. It is necessary to clarify which aspects are considered in the review.

2) Weak categorization of the article: the article contains three sections - 1, 2 and 3.3. Most likely, 3.3 is a typo. Section 2 needs to be broken down into subsections (eg, VUV, plasma, etc.).

3) From my point of view, the article consideres an insufficient amount of literature - 28 articles, of which 5 - with the author of the article. The volume of the analyzed literature should be at least doubled.

Author Response

Dear Reviewer, thank you for your comments.

The article presents an overview of modern strategies to improve PEEK wettability and adhesion. Although the review is up-to-date, here are some comments on it:

1) The abstract should be expanded. It is necessary to clarify which aspects are considered in the review.

Thanks for this comment. I expanded the abstract with more details and stressed the crucial parameters that enable optimal wettability. The abstract is now: “PEEK is the material of choice in several applications ranging from the automotive industry to medicine, but the surface properties are usually not adequate. A standard method for tailoring surface properties is the application of gaseous plasma. The surface finish depends enormously on the processing parameters. This article represents a review of strategies adapted for improved wettability and adhesion of PEEK. The kinetics of positively charged ions, neutral reactive plasma species, and the vacuum ultraviolet radiation on the surface finish are analyzed, and synergies are stressed where appropriate. The reviewed articles are critically assessed regarding the plasma and surface kinetics, and the surface mechanisms are illustrated. The directions for obtaining optimal surface finish are provided together with the scientific explanation of the limitations of various approaches. Super-hydrophilic surface finish is achievable by treatment with a large dose of vacuum ultraviolet radiation in the presence of oxidizing gas. Bombardement with positively charged ions of kinetic energy between about 100 and 1000 eV will also enable high wettability, but one should beware of excessive heating when using the ions.”

2) Weak categorization of the article: the article contains three sections - 1, 2 and 3.3. Most likely, 3.3 is a typo. Section 2 needs to be broken down into subsections (eg, VUV, plasma, etc.).

Thanks for spotting this typo. To address the categorization, I broke section 2 into four subsections: 2.1 Conventional oxygen plasma, 2.2 VUV radiation, 2.3 Energetic ion beams, 2.4 Plasmas in other gases or gas mixtures

3) From my point of view, the article consideres an insufficient amount of literature - 28 articles, of which 5 - with the author of the article. The volume of the analyzed literature should be at least doubled.

This is also a recommendation of Referee #1. I wanted to keep the article as concise as possible. In fact, Referee #1 found the title inappropriate, and I agree. So I changed the title to focus on non-equilibrium plasma treatments to improve wettability. I reexamined my self-citations and found them necessary for explaining the results. I examined some additional literature and explained the observations reported by various authors.

Kind regards

Round 2

Reviewer 1 Report

Author has revised the manuscript carefully according to the reviewer’s comments. It can be accepted under this current version.

Reviewer 3 Report

The article has been revised and can be accepted for publication in present form